# Effects of Transcranial Pulse Stimulation (TPS) on Adults with Symptoms of Depression—A Pilot Randomized Controlled Trial

**DOI:** 10.3390/ijerph20032333

**Published:** 2023-01-28

**Authors:** Teris Cheung, Tim Man Ho Li, Yuen Shan Ho, Georg Kranz, Kenneth N. K. Fong, Sau Fong Leung, Simon Ching Lam, Wing Fai Yeung, Joyce Yuen Ting Lam, Kwan Hin Fong, Roland Beisteiner, Yu-Tao Xiang, Calvin Pak Wing Cheng

**Affiliations:** 1School of Nursing, The Hong Kong Polytechnic University, Hong Kong SAR, China; 2Department of Psychiatry, The Chinese University of Hong Kong, Hong Kong SAR, China; 3Department of Rehabilitation Sciences, The Hong Kong Polytechnic University, Hong Kong SAR, China; 4School of Nursing, Tung Wah College, Hong Kong SAR, China; 5Department of Neurology, Medical University of Vienna, 1090 Vienna, Austria; 6Unit of Psychiatry, Department of Public Health and Medicinal Administration, Institute of Translational Medicine, University of Macau, Macao SAR, China; 7Department of Psychiatry, University of Hong Kong, Hong Kong SAR, China

**Keywords:** transcranial pulse stimulation, noninvasive brain stimulation, efficacy, major depressive disorder

## Abstract

Transcranial pulse stimulation (TPS) is a recent development in non-invasive brain stimulations (NIBS) that has been proven to be effective in terms of significantly improving Alzheimer patients’ cognition, memory, and execution functions. Nonetheless, there is, currently, no trial evaluating the efficacy of TPS on adults with major depression disorder (MDD) nationwide. In this single-blinded, randomized controlled trial, a 2-week TPS treatment comprising six 30 min TPS sessions were administered to participants. Participants were randomized into either the TPS group or the Waitlist Control (WC) group, stratified by gender and age according to a 1:1 ratio. Our primary outcome was evaluated by the Hamilton depression rating scale-17 (HDRS-17). We recruited 30 participants that were aged between 18 and 54 years, predominantly female (73%), and ethnic Chinese from 1 August to 31 October 2021. Moreover, there was a significant group x time interaction (F(1, 28) = 18.8, *p* < 0.001). Further, when compared with the WC group, there was a significant reduction in the depressive symptom severity in the TPS group (mean difference = −6.60, *p* = 0.02, and Cohen’s *d* = −0.93). The results showed a significant intervention effect; in addition, the effect was large and sustainable at the 3-month follow-up. In this trial, it was found that TPS is effective in reducing depressive symptoms among adults with MDD.

## 1. Introduction

Since the emergence of the COVID-19 pandemic, there has been a great deal of published work examining the associations between COVID-19 and mental health. In particular, there have been studies conducted globally on anxiety and depression in the general population [1,2,3], as well as in vulnerable subpopulations, which include frontline nurses [4], healthcare workers, [5] and older adults [6]. 

A large-scale multi-country cross-sectional study was conducted involving ten countries comprising 25,053 individuals between 24 March and 30 April 2020 [3]. This study aimed at examining the psychosocial wellbeing of the general population in different countries amidst the COVID-19 pandemic. Hong Kong was one of the regions in Asia that was investigated in this study [3]. Of particular note was that the prevalence of depression among Hong Kong citizens was 45.7% (*n* = 5254). Hong Kong also ranked as having the highest prevalence rate of depression among the five Asian countries/regions (China, Macau, the Republic of Korea, and the Philippines making up the others) that were investigated in this multi-country study. Despite the rapid increase in depression prevalence in Hong Kong, there seems to be a lack of robust, large-scale, and technology-based interventional studies—both local or nationwide—in regard to restoration of the general public’s optimal psychosocial wellbeing amidst the COVID-19 pandemic.

Depression is a debilitating disorder affecting individuals’ levels of bio-psychosocial functioning across different age groups [3]. Without timely mental health research [7] and early psychosocial intervention, Hong Kong may soon develop a depression epidemic, which may, in turn, increase the global disease burden and healthcare expenditure. Thus, there is a pressing need to formulate effective interventions in order to mitigate the negative detrimental impact brought by the COVID-19 pandemic [7].

### 1.1. Conventional Treatment Approach towards Depression in Psychiatry

Traditionally, pharmacological treatment, physical treatment, and psychotherapy have been used as conventional treatments in psychiatry. Pharmacological treatment includes the prescription of anti-depressants, such as tricyclic antidepressants (TCAs), selective serotonin reuptake inhibitors (SSRIs), and serotonin–norepinephrine reuptake inhibitors (SNRIs) [8]. In regard to physical treatment, electroconvulsive therapy is a conventional approach [9]. All these approaches could increase neuroplasticity and lead to release of brain-derived neurotrophic factors (BDNF). Nonetheless, biological treatments alone may not be sufficient to treat all patients with depression, especially if some individuals do not have the normal ability to adapt through neuroplasticity. Thus, psychotherapy (e.g., cognitive behavioral therapy) is another common treatment option concurrently used in order to induce clinical improvement in depressive symptoms [10]. 

### 1.2. Non-Intrusive Brain Stimulation (NIBS) 

In the last decade, there have been some other treatment approaches using non-intrusive brain stimulation (NIBS), such as transcranial direct current stimulation (tDCS) [11] and transcranial magnetic stimulation (TMS) [12,13], as well as repetitive transcranial magnetic stimulation (rTMS) [14], in the treatment of depression. The recent development of a new type of NIBS called **‘transcranial pulse stimulation’** (TPS), also known as low-intensity extracorporeal shock wave therapy (Li-ESWT), has been proven effective for only a 2-week treatment of 35 patients with Alzheimer’s disease (AD). Patients’ cognition and memory showed significant improvement, which lasted up to 3 months [15]. However, there is a lack of scientific evidence on the efficacy of this TPS intervention on other psychiatric populations, such as those with major depressive disorder (MDD), which is increasingly prevalent in Hong Kong and nationwide, especially during the COVID-19 pandemic [2,3,16].

### 1.3. How Does TPS Work on the Human Brain?

TPS uses repetitive, single, ultrashort pulses in the ultrasound frequency range in order to stimulate the brain. With a neuro-navigation device, TPS can target the human brain in a highly focal and precise manner [15]. TPS differs from transcranial direct current stimulation (tDCS) and transcranial magnetic stimulation (TMS), as these two variations use direct or induced electric currents. Using an electric current to stimulate the brain may be limited by the problem of conductivity [17] and a potential failure to reach deep brain regions [18]. TPS, however, uses low-intensity, focused ultrasound, which provides good spatial precision and resolution that can better modulate subcortical areas noninvasively, in spite of the problem of skull attenuation [19]. By utilizing lower ultrasound frequencies, TPS can successfully improve skull penetration in the human brain [15].

### 1.4. Biological Mechanism of TPS

The basic mechanism of TPS is found in the process of mechanotransduction. TPS can stimulate vascular growth factors (VEGF) [20,21] and brain-derived neurotrophic factors (BDNF) [22], improve cerebral blood flow, promote the formation of new blood vessels (angiogenesis), and also promote nerve regeneration. It is a biological pathway through which the cells convert the mechanical TPS stimulus into biochemical responses, thus influencing some fundamental cell functions such as migration, proliferation, differentiation, and apoptosis [23,24]. TPS can stimulate deep cerebral regions, reaching as far as 8 cm into the brain. The ultrashort ultrasound pulse could enhance the cell proliferation and differentiation in cultured neural stem cells, which play an important role in the repair of brain functions in central-nervous-system-based diseases [25]. Moreover, TPS may affect neurons and induce neuroplastic effects through several pathways, resulting in an increase in cell permeability [25], stimulation of mechanosensitive ion channels [24], the release of nitric oxide (which may lead to vasodilation), increased metabolic activity, and angiogenesis [26]. Research has proven that the serum levels of BDNF were reduced in patients diagnosed with major depressive disorder [27], which indicated that BDNF played a critical role in the pathophysiology of depression. As such, it can be said that a reduced production of BDNF and neuroplasticity can lead to depression [28].

### 1.5. Clinical Effects of TPS

Focused ultrasound demonstrated the neuromodulation effect in the human brain. This is because it can modulate the amplitude of somatosensory evoked potentials (SEPs) (when targeted at the cortical regions that generate these potentials) [29] and can even modulate the parts of the deep structure, such as the thalamus [19]. 

### 1.6. Past Research on TPS

In the last decade, there were two studies that utilized TPS in order to treat the disease population. In 2014, five patients with unresponsive wakefulness syndrome received a 4-week TPS treatment (three times per week), 4000 pulses each, every 6 months for an average of 2 to 4 years. Patients showed significant improvement in vigilance, and three patients’ percutaneous endoscopic gastrostomy (PEG) tube could be removed due to their improved oropharyngeal motor function [30]. 

In another recent study, 35 older adults with Alzheimer’s disease (AD) were treated in three TPS sessions (6000 pulses each) per week for 2–4 weeks, either over classical AD-affected sites (such as the dorsolateral prefrontal cortex, areas of the memory, and language network) or over all accessible brain areas (i.e., global brain stimulation). Results showed significant improvement in the CERAD (Consortium to Establish a Registry for Alzheimer’s Disease) score, immediately after intervention and at 1 month and 3 months after intervention. The results from fMRI also showed significantly increased connectivity within the memory network [31]. In addition, Beisteiner’s study also found that participants’ depressive symptoms were significantly improved—as measured via the geriatric depression scale (GDS) (*p* = 0.005) and Beck depression inventory (BDI) (*p* < 0.0001) at 1 month and 3 months post-stimulation follow-up when compared with the baseline scores [31]. 

In regard to the GDS, the effect of TIME was significant (*p* = 0.005). Pairwise comparisons (conducted via the Wilcoxon test) showed GDS improvement for baseline >3 months post stimulation (PBonf = 0.012). In regard to BDI, the effect of TIME was also significant (*p* < 0.0001). Pairwise comparisons (Wilcoxon test) displayed BDI improvements for baseline > post-stimulation (PBonf = 0.012), baseline > 1 month post stimulation (PBonf = 0.006) and baseline > 3 months post stimulation (PBonf = 0.012). Importantly, there was no significant correlation between the BDI/GDS scores and the global CERAD scores (CTS, LR), or in the PCA factors after accounting for multiple comparisons (i.e., through Bonferroni corrections). This indicates that CERAD improvements were not driven by changes in depressive symptoms [31].

### 1.7. Research Gap

Ultrasound for the brain is a revolutionary therapeutic treatment approach in patients with neuropsychiatric symptoms. As TPS is a relatively new NIBS technology, there have been only a few studies regarding its use; further, these studies were all conducted on older adults with mild neurocognitive disorders [31]. There is, currently, no study that has been conducted on adults with depression in Hong Kong or nationwide. This fact provides us with the impetus to execute the current study in order to bridge the research gap as well as to evaluate the efficacy of TPS in treating the depression among adults in Hong Kong.

To the best of our knowledge, our study is the first pilot, randomized, controlled trial (RCT) conducted nationwide to use TPS to treat adults with major depressive disorder (MDD) in Hong Kong. Health policymakers and researchers are working to ameliorate hard-to-formulate effective mental health interventions in order to curb the depression epidemic brought on by the pandemic [32] and beyond. It is envisaged that our findings can be transferable to other cultural contexts. The data emerging from this study will facilitate cross-cultural and interdisciplinary collaboration. It will be able to provide an evidence-based treatment for individuals with MDD. Our findings may contribute new knowledge on the efficacy of using TPS as a NIBS treatment of MDD. It also lays down the groundwork for a larger clinical trial in the near future, as well as the basis to evaluate if this new treatment modality can be used to treat patients suffering from other types of neurodegenerative diseases or neuropsychiatric disorders.

### 1.8. Objectives of This Study

The primary objective of this study was to evaluate the effects of transcranial pulse stimulation (TPS) on participants’ depression severity scores among adults in Hong Kong. The secondary objectives included examining the effects of TPS on participants’ anhedonia symptoms, and instrumental activities of daily living (IADL), and cognition and to examine if there were any brain functional connectivity changes in the participants’ brains after TPS. Nonetheless, in this article, we have chosen to only focus on the effects of TPS via psychological instrumental scores; we will endeavor to report structural and functional connectivity changes elsewhere. 

## 2. Materials and Methods

### 2.1. Study Design

In this study, we used a single-blind randomized controlled trial design with two-armed repeated measures. The trial design complied with the Consolidated Standards of Reporting Trials (CONSORT) statement [33] and the Good Clinical Practice guidelines. In our two-armed design, we used TPS as an intervention group and a waitlist control (WC) group. A WC group was appropriate for the purposes of comparing the effect of the TPS on the intervention group to that of those who did not receive the TPS treatment at the same timepoints [34]. Both groups were measured at baseline (T1), immediately after the intervention (T2), and at the 3-month follow-up (T3) (refer to Figure 1 CONSORT flow diagram). 

### 2.2. Sample Size Estimation

To the best of our knowledge, there have been only two studies using TPS on a disease group. Of these, one study was an uncontrolled pilot study conducted on 35 patients with AD in Austria [15], while the other study’s sample size was too small [30]; therefore, we cannot compare their effect size with our estimated sample size in this study. Considering the nature of our study as the first pilot RCT evaluating the efficacy of TPS in the treatment of depression, we thus referenced Beisteiner’s TPS study [15] and aimed at recruiting 30 participants in this study.

### 2.3. Subject Recruitment

Participants were recruited via a mass email invitation delivered to the eligible subjects via collaborators in the Hong Kong Polytechnic University (HKPU) and University of Hong Kong (HKU). A QR code flyer embedded with an electronic application form was flagged up in all communal areas in HKPU and HKU campus. The recruitment period spanned for three months from 1 August to 31 October 2021.

### 2.4. Subjects

The inclusion criteria for participants were as follows (in order to obtain a homogenous sample): (1) must be aged 18 or over; (2) must be able to understand/read Chinese; (3) must possess a HAM-D-17 score of ≥8; and (4) must be able to provide written informed consent. All subjects undertook the screening procedures to ensure the eligibility.

Exclusion criteria included individuals who (1) had a DSM-5 diagnosis other than major depressive disorder (e.g., bipolar affective disorder or schizophrenia); (2) possessed an alcohol or substance dependence; (3) had a concomitant unstable major medical conditions or major neurological conditions, such as brain tumor, brain aneurysm, etc.; (4) had hemophilia or other blood clotting disorders or thrombosis; (5) possessed significant communicative impairments; (6) possessed a metal implant in the brain or in a treated area of the head; (7) undertook corticosteroid treatment within the last six weeks before the first TPS treatment; and (8) were pregnant or breastfeeding women.

Prior to participation, the principal investigator had conducted a phone interview with all eligible subjects enquiring about their physical and mental health history to ensure that no subjects with identifiable risks were included in this trial.

### 2.5. Randomization, Allocation, and Masking

All eligible participants were listed according to their surnames in alphabetical order; additionally, each participant was assigned a unique identifier. An off-site independent statistician used a computer-generated list of random numbers (www.random.org) (accessed on 5 November 2021) to ensure concealment of randomization. After obtaining participants’ baseline measurement in regard to their HAM-D-17 score, randomization was conducted by an independent statistician off-site using a stochastic minimization program [35] in order to balance the gender, age, and HAM-D-17 scores of the participants. The eligible participants were randomized into either the TPS group or the WC group, on a 1:1 ratio. Participants in both groups were blinded about their grouping in order to minimize potential contamination of the effects of TPS or subject bias.

### 2.6. Intervention (Transcranial Pulse Stimulation)

The TPS intervention was performed at the Integrative Health Clinic in the Hong Kong Polytechnic University. Two licensed medical professionals delivered the intervention.

#### 2.6.1. TPS Procedures

The TPS system consists of a mobile single transducer and an infrared camera system which incorporates neuro-navigation. This system was developed by NEUROLITH, Storz Medical AG, Tägerwilen, Switzerland. In addition, this TPS system can generate single ultrashort (3 µs) ultrasound pulses with 0.2–0.25 energy levels (mJ/mm^2^) and 4–5 Hz pulse frequencies (pulses per second). During the TPS session, participants sit in an adjustable electronic chair and wear a BodyTrack system, which consists of a tracking glass with markers, a 3D camera, and a TPS handpiece (Figure 2). The reason for wearing this BodyTrack system was to ensure that the participant’s head matched with their own fMRI (T1 images), such that the interventionist could visualize each pulse applied and document it in real time. The advantage of the real-time tracking of the handpiece position is that it enabled the automatic visualization of the treated brain region and highlighted it in green color (Figure 3).

The interventionist used the TPS handpiece over the participants’ skull, such that the participants’ fMRI T1 brain images could be visualized and projected in real-time. The TPS treatment was performed by a licensed mental health professional holding the applicator at hand. Each TPS session was recorded for the purposes of post-TPS evaluation of intracerebral pulse localizations (Figure 4). 

In this trial, we targeted the stimulation to the left dorsal lateral prefrontal cortex (DLPFC), guided by a real-time MRI brain imaging. The basis for the choice of the selected brain region was based on previous research [36] into depression. Specifically, it was due to the fact that there is an imbalance between left and right DLPFC in patients with major depressive disorder (MDD). Patients with MDD have been associated with hypoactivity in the left DLPFC, thereby leading to negative emotional judgments. Past research has proven that stimulation of the left DLPFC can effectively improve the depressed mood with different non-invasive brain stimulation techniques, such as TMS and tDCS [37].

#### 2.6.2. Intervention Dose

Each participant undertook the pre-treatment fMRI scan, which was performed in the University Research Facility in Behavioural and Systems Neuroscience at the Hong Kong Polytechnic University, Hong Kong, in order to ensure that participants’ brains had no structural defects, tumors, brain trauma, or any other brain abnormalities. In this study, we delivered 300 pulses to the subjects’ left DLPFCs in each session (total: 1800 pulses). All participants (both the TPS group and the WC group) received six 30 min TPS sessions with three sessions per week on alternate days for two consecutive weeks. The energy levels ranged from 0.2–0.25 mJ/mm^2^ and a pulse frequency of 3–4 Hz was used. 

To ensure the fidelity of the intervention, the project team ascertained whether the interventions were delivered as intended in the study protocol. The interventionist (PI) possesses a PhD in the Social Sciences (HKU) and is a UK- and HK-licensed mental health professional with more than 10 years of clinical experience in mental health and neuroscience. The research associate issued WhatsApp message reminders (e.g., TPS intervention schedule, fMRI scan appointments, and f/u appointments slips) to subjects, monitoring the subjects’ progress, any adverse effects, and also adherence throughout the trial period.

### 2.7. Ethical Considerations

All participants were required to provide written informed consent in order to participate in this study. Ethical approval was sought from the Human Subjects Ethics Sub-committee, and research safety approval was also obtained prior to commencement of the study within the Hong Kong Polytechnic University (reference #: HSEARS20210608002). This study adhered strictly to the ethical principles detailed in the Declaration of Helsinki that was developed by the World Medical Association. Potential risks involved in fMRI and other potential side effects of TPS were clearly indicated in the information sheet. Voluntary participation, anonymity, confidentiality, and the right to withdraw were all respected. All participants were given a HKD 50 supermarket cash coupon for their time, as well as travel-related expenses for the 3-month follow-up period. This trial is registered with Clin.Trials.gov (ref: NCT05006365).

### 2.8. Baseline Assessments

Each participant was asked to produce their prescribed drug formulation sheet, endorsed by their private/public psychiatrists in order to confirm their psychiatric diagnosis of MDD. Diagnosis of major depressive disorder (MDD) was confirmed using the structured clinical interview for diagnostic and statistical manual of mental disorders (SCID-5). 

Demographic data including age, gender, educational attainment, marital status, employment status, occupation, duration of possessing MDD, drug-taking (years/months), and family household income were all solicited.

### 2.9. Outcome Measurements

The primary outcome (i.e., depression) was assessed by the 17-item Hamilton rating scale for depression (HDRS-17) [38] in order to measure symptoms of depression and the participants’ mood. HDRS-17 is a widely used reliable measurement of depressive and mood symptoms. Scores ranged from 0 and 52, with higher scores indicating more severe depression. A clinical response was defined as a reduction of 50% or more in the HDRS-17 score. A HDRS-17 score of ≤7 was used as an indicator of remission.

Regarding the secondary outcomes, anhedonia was assessed by the Chinese version of the Snaith–Hamilton pleasure scale (SHAPS) (the clinical utility of the Snaith–Hamilton pleasure scale was utilized through the Chinese settings). This scale is one of the most widely used self-report questionnaires in clinical research. It is used for the purposes of evaluating anhedonia with good psychometric properties [39]. The instrumental activities of daily living (IADL) was assessed by the Hong Kong Chinese version of the Lawton instrumental activities of daily living scale. This version of the IADL is a valid and reliable tool to use in order to assess the daily functioning of Hong Kong adults [40]. In addition, cognition was measured via the Hong Kong Chinese version of the Montreal cognitive assessment (MoCA) [41]. Working memory, executive function, and attention were all measured by a forward and backward digit span, as well as the trail making test-A and B.

### 2.10. Safety Issues, Adverse Effects, and Risk Indicators of TPS

TPS uses very low energy for brain stimulation. An in vivo animal study showed that the performance of TPS did not cause any tissue damage despite using 6–7-fold higher energy levels when compared with those used in human studies. Furthermore, the intervention did not cause any serious adverse effects, such as intracranial bleeding, oedema, or other types of intracranial pathology, as confirmed via magnetic resonance imaging (MRI) in a previous AD study. Only a few subjects reported headache (4%), pain or pressure (1%), and/or mood deterioration (3%) [15]. The clinically certified (CE)-marked TPS system has proven to be safe in >1500 treatments. Nevertheless, we prepared a checklist stating all the potential adverse effects associated with TPS administration [31]. This checklist was used in order to monitor subjects’ tolerability, as well as any adverse events that may occur in each session, all throughout the trial period. In our study, we found that a few subjects reported headaches (4%). However, no subject required ingesting any type of pain analgesics for their headaches. Moreover, only one subject complained of nausea and vomiting after the first TPS treatment. However, symptoms subsided within 2 h and no such complaint was made thereafter.

### 2.11. Statistical Analysis

All statistical analyses were performed using statistical software R for Windows (R version 4.1.0). Means and standard deviations (SD) for the continuous variables are presented, while numbers and percentages for the categorical variables are shown. A *p*-value <0.05 was considered statistically significant. Sociodemographic differences between the intervention group (TPS group) and the waitlist control (WC) group were analyzed using chi-square tests and *t*-tests. If significant differences between sociodemographic factors were found, covariates were considered as confounding variables in the analyses.

The normality of the primary outcome (i.e., depression) scores was tested by the Shapiro–Wilk test for each combination of the factor levels (group and time). The *t*-test was used to test the baseline difference. An analysis of variance (ANOVA) with repeated measures was utilized in order to test the group (between-subject factor), time (within-subject factor), and the group x time interaction effects of the primary outcome between the TPS groups and the WC group. A significant interaction indicates that the effect of the TPS intervention on the primary outcome depends on time. Post hoc comparisons between groups and time points were conducted using a *t*-test with the Bonferroni correction. 

The normality of the secondary outcome scores was tested by the Shapiro–Wilk test for each time point. For normally distributed outcomes, ANOVA with repeated measures was used to determine whether the outcome scores were significantly different between pre- and post-test. For the outcome scores that were not normally distributed, a non-parametric Friedman test was used in order to test the mean difference. A Cohen’s *d* effect size (ES) for each outcome was calculated, where *d* = 0.2, 0.5, and 0.8 correspond to small, medium, and large effect size, respectively [42]. 

## 3. Results

### 3.1. Sociodemographic Characteristics between the TPS Group and the Waitlist Control (WC) Group

Table 1 shows the sociodemographic differences between the TPS group and the WC group. There were no statistically significant differences between the two groups. Thus, subsequent analyses were not adjusted for any sociodemographic factors. More females (73%) participated in this study than males (27%). The mean age of this cohort was 38.8 (SD 15.0) and 34.4 (SD 16.5) for the TPS group and WC group, respectively. Only one participant was divorced, while others were single, married, or in a relationship. More than one third (36.7%) were full-time students, 13.3% were licensed professionals, and the remaining half of the participants were managerial/administrative/clerical workers, housewives, retirees, or unemployed. Due to the diversity of participants’ employment, income dispersion was noted between licensed and non-licensed participants. The majority of the participants (80%) had completed tertiary education. A total of 93% of participants did not have a chronic illness. All participants had a clinical diagnosis of major depressive disorder (MDD). The mean duration of suffering from MDD was over 8.1 years and 4 years for the TPS group and WC group, respectively. Interestingly, participants reported taking prescribed antidepressants (selective serotonin reuptake inhibitors (SSRIs)) for only 2.79 years and 3.26 years for the TPS group and WC group, respectively. A total of seven participants were not taking any prescribed antidepressants (due to fear of the undesirable side effects of the drugs and/or drug tolerance). Other sociodemographic factors were distributed equally among their categories. All of the 30 participants, whether in the TPS group or the WC group, completed six TPS sessions, and 28 participants attended the 3-month follow-up. Hence, the attrition rate was 6.7% in this trial.

### 3.2. Effect of the TPS Intervention on Depression

Table 2 reports the effect of the intervention. As can be seen from the data, there was no baseline difference between the TPS group and the WC group. The primary outcome (i.e., depression) score was normally distributed in each group at each time point (*p* > 0.05). There was a significant group x time interaction, F(1, 28) = 18.8, *p* < 0.001. The mean depression score was significantly different between the TPS group and the WC group at post-test (*p* = 0.02) but not significant at pre-test (*p* = 0.15). There was a significant intervention effect, with a large effect size of −0.93.

### 3.3. Differences in Secondary Outcomes between Baseline and Post-Test

Table 3 demonstrates the differences in the secondary outcomes between pre- and post-test. The secondary outcome scores were not normally distributed at all time points (*p* < 0.05), except for the anhedonia score (*p* > 0.05). There were significant time effects on the secondary outcomes. The effect on cognition (*d* = 0.88) was large, whereas the effects on anhedonia (*d* = −0.79) and digit span backward (*d* = 0.69) were medium. The effects on the trail making test-A (*d* = −0.40), IADL (*d*=0.40), and digit span forward (*d* = 0.43) were small. There was no significant effect on TMT B (*p* > 0.05). 

### 3.4. Differences in Primary and Secondary Outcomes between Baseline, Post-Test, and at the 3-Month Follow-Up

Table 4 reports the effects of the TPS intervention from baseline to post-test and at the 3-month follow-up. Our findings showed significant results leading to a further reduction in depression severity; in addition, the effect size for depression was large (d = 1.35). The effect sizes for all other secondary outcomes (i.e., global cognition, TMT B, and digit span backward) were very large (d = 1.2, 1.1, and 1.09, respectively). The effects analyzed in the trail making test-A, IADL, SHAPS, and digits span forward were medium to large (d = 0.5–0.8). With the exception of digit span forward (*p* = 0.2), all the primary and secondary outcomes were highly significant (all *p* < 0.001).

## 4. Discussion

This is the first nationwide pilot RCT that evaluated the effects of TPS on adults with MDD in Hong Kong. Although the mechanism of TMS, tDCS, and TPS are different, previous studies have proven that stimulating the DLPFC region was effective. In this trial, we targeted the DLPFC in order to evaluate the effects of TPS on depression. Our results show that participants’ immediate post-stimulation scores showed a significant reduction in depression symptoms and the effect size (ES) was very large (Cohen’s d = 0.9). More importantly, the effect of TPS is sustainable at the 3-month follow-up period (Cohen’s *d* = 1.35). Our findings echoed another study [37] that used TMS and tDCS in order to treat the same brain region, leading to a significant improvement in depressive symptoms.

### 4.1. Cognition

Cognition is commonly affected in brain disorders. In this regard, MDD is no exception. However, NIBS may have procognitive effects, with a high tolerability. A meta-analysis [43] was conducted that evaluated the efficacy of transcranial magnetic stimulation (TMS), as well as transcranial direct current stimulation (tDCS). The study investigated the factors in improving cognition in different neuropsychiatric disorders, including schizophrenia, depression, dementia, Parkinson’s disease, stroke, traumatic brain injury, and multiple sclerosis. A total of 82 studies (n = 2784) were included in this meta-analysis. Nonetheless, both TMS (ES = 0.17 and *p* = 0.015) and tDCS (ES = 0.17 and *p* = 0.021) showed significant but small trans-diagnostic effects on working memory. Nevertheless, in our study, we found that TPS can significantly and progressively improve participants’ cognition when comparing with their baseline scores (Table 4). Moreover, the ES was large at post-test (Cohen’s *d* = 0.88) to very large at the 3-month follow-up (Cohen’s *d* = 1.2). In other words, TPS seemed to be more effective and sustainable in improving cognition in MDD patients than existing TMS or tDCS. Such a discrepancy in the efficacy between TMS/tDCS and TPS may be attributed to the fact that TPS can stimulate deep cerebral regions, reaching as much as 8 cm into the brain, and, thus, TPS should have, to a large extent, a reduced individual brain/skull conductivity problem [17].

### 4.2. Anhedonia

Regarding the clinical manifestation, anhedonia is one of the core symptoms of depression [44] but it remains difficult to treat [45]. Cognitive abnormalities are also a core feature of depression, and involve attention, memory, executive functions, and psychomotor speed [46]. Impaired cognition, emotional processing, and psychosocial dysfunction are the primary causes of psychosocial dysfunction in depression [47]. Nonetheless, research has proven that improvement in anhedonia is positively correlated with improvement in psychosocial functioning among depressive individuals [48]. Our study findings have affirmed existing research findings that improvement in depressive symptoms may lead to improved cognition, attention, memory, and executive function [46] and reduced anhedonia symptoms [44]. Although TMS has been proven as an effective treatment for depression, there is a lack of consensus on whether anhedonia can be used as a predictive biomarker of MDD. A previous study [45] applied TMS on 144 community-dwelling out-patients with depression. Results showed a significant improvement in anhedonia from pre- to post-treatment (7.69 ± 3.88 vs. 2.96 ± 3.45; *p* < 0.001). Significant correlations between improvements in anhedonia and other depressive symptoms were also noted (*r* = 0.55 and *p* < 0.001). Our study used the same instrument (SNAPS) to measure anhedonia, with significant improvement at 2-weeks post-stimulation (*p* < 0.001, *d* =-0.79) and at the 3-month follow-up (*p* <.001, *d* = −0.81). It indicated that depressive symptoms are also positively and significantly correlated with anhedonia. In this regard, our findings echoed existing findings in that reduction in depressive symptoms may lead to reduced anhedonia [44].

### 4.3. Instrumental Activities of Daily Living (IADL), Working Memory, and Executive Function

Our results show that participants have significant improvement in the IADL (*p* < 0.001 and Cohen’s *d* = 0.4), working memory, and executive functions as revealed by participants’ significant improvement in the working memory scores of the trail making test-A (TMT-A) (*p* < 0.001, Cohen’s *d* = −0.4) and digit span (DS) (Forward and Backward, *p* < 0.05, Cohen’s *d* = 0.43 and 0.69, respectively). This was despite the effect of TPS on IADL, TMT-A, and DS being medium at the 2-week post-stimulation. Nonetheless, IADL, TMT-A, and DS (Forward and Backward) remained highly significant at the 3-month follow-up (all *p* < 0.001), though the effect was somewhat similar in IADL (*d* = 0.7), DS Forward (*d* = 0.54), and TMT-A (*d* = −0.64). However, it must be noted that DS Backward had a very large effect at the 3-month follow-up (*d* = 1.09). Our findings prove that TPS can not only improve participants’ instrumental activities of daily living but can also improve their working memory and executive function.

Our findings were echoed by past research conducted by a previous study [49] that used TMS in order to interrogate the effects of activities of daily living (ADL) on 62 stroke patients in a randomized sham-controlled trial. Results showed that there was a significant difference in the functional ADL (13.00 SD 1.69 vs. 4.21 SD 2.96) and improved attention in the trail making test-A (96.67 SD 25.18 vs. 44.28 SD 19.45) and digit span test (96.67 SD 25.18 vs. 44.28 SD 19.45) in the post-TMS treatment group when compared with the sham group.

In this trial, we demonstrated that TPS significantly reduces depressive symptom severity, improves IADL and global cognition, reduces anhedonia, and improves working memory and executive function through just two weeks’ time of application. Further, the effects are shown to be sustainable at a 3-month follow-up. Although TPS is a novel pulsed ultrasound technique, it may well represent an add-on therapy that can be applied concurrently with other existing treatment approaches in psychiatry [50]. In summary, TPS is a safe and effective top-on treatment modality that can be used to treat symptoms of depression, especially when compared with existing NIBS technology in psychiatry.

### 4.4. Limitations of This Study

We executed the first pilot RCT nationwide which interrogated the effects of TPS on MDD. We can only, thus, use existing NIBS studies, such as on TMS and tDCS, to act as comparison for our findings. As both TMS and tDCS use non-invasive neuromodulation technology via electrical stimulation, the mechanism is still different from TPS, which uses ultrashort ultrasound waves in its stimulation of the human brain. Thus, our findings need to be interpreted with caution. In addition, there was only one uncontrolled pilot study that used fMRI, which confirmed the significant effects of TPS on cognition, memory, and executive function of patients with Alzheimer’s disease [15], while our RCT results were derived from self-report data. Of particular note is the fact that participants in our waitlist control group were given the same TPS intervention, albeit in different timepoints, and they all received the same TPS pulse/frequency/energy in this TPS trial. In addition, the TPS machine itself may have potential placebo effects that may influence the results in terms of potential bias. In addition, there could also be a potential risk of contamination if our WC Group utilized self-prescribed pharmaceutical drugs or other TCM modalities (e.g., herbal soup, acupuncture, etc.) to treat their symptoms of depression whilst waiting for their turn for the TPS treatment. Such contamination may impact the receptivity and efficacy of our TPS intervention. Additionally, two subjects attrited from the 3-month post-stimulation follow-up, which left 28 subjects for evaluating the long-term sustainability of the TPS on depression. Due to budget and time constraints, we could not propose a larger sample size in the protocol. Given the fact that this was a pilot study, we recommend future replication of a larger-scale RCT using a double-blinded sham-controlled study in the future. Last but not least, individuals with severe depression, i.e., those with fleeting suicidal ideas/plans, were excluded in this study. Thus, the efficacy of TPS on these severely depressive community-dwelling individuals remains unknown. 

## 5. Conclusions

TPS is the latest technological NIBS device that has been proven effective, safe, and sustainable for reducing depressive symptom severity in this pilot RCT. The utilization of NIBS to treat neurodegenerative diseases or neuropsychiatric symptoms is likely to be a future trend in neuroscience and clinical psychiatry. In this vein, TPS may well be considered a top-on treatment option, especially for treatment-resistant patients and those who are seeking prompt recovery. Future replication of a multi-center study may wish to adopt a double-blinded, sham-controlled RCT in order to filter out the placebo effects on a larger sample.

## Figures and Tables

**Figure 1 ijerph-20-02333-f001:**
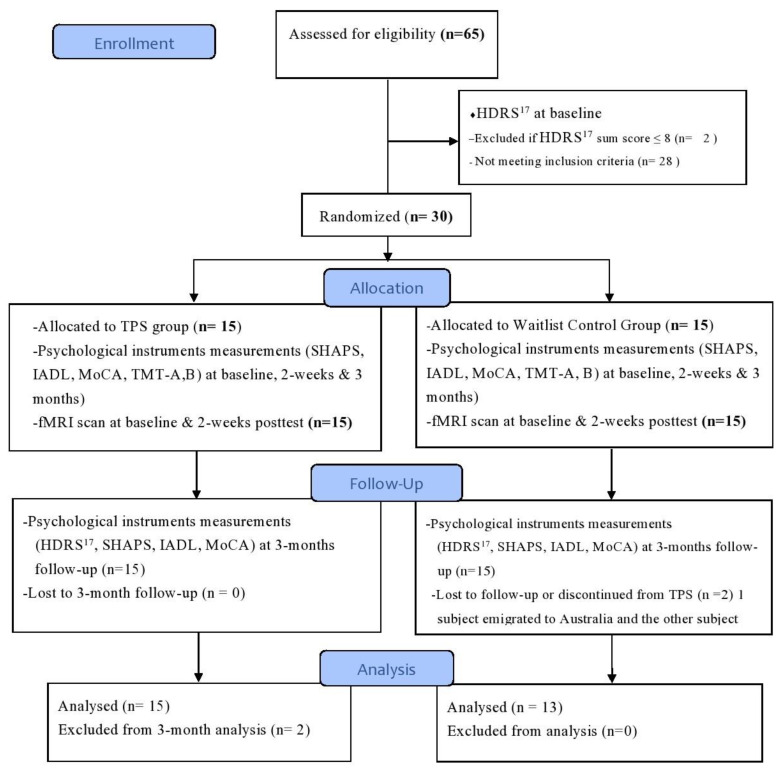
CONSORT flow diagram.

**Figure 2 ijerph-20-02333-f002:**
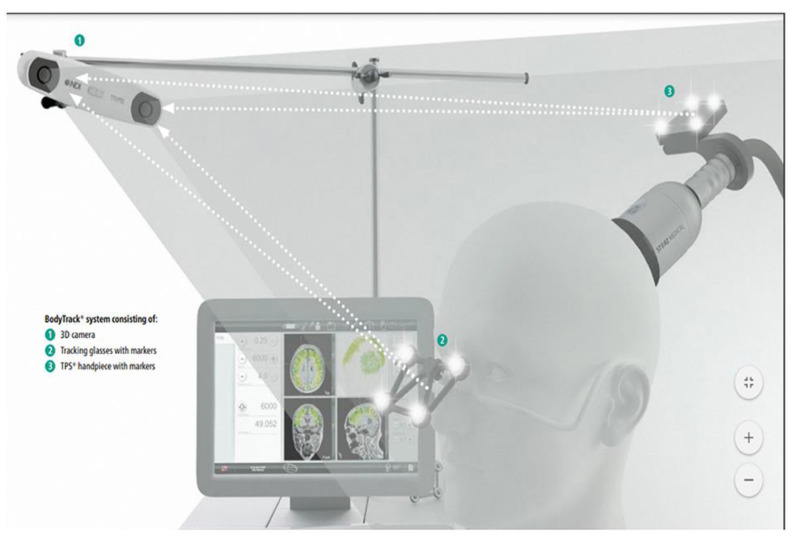
Courtesy image from NEUROLITH—TPS MANUFACTURER.

**Figure 3 ijerph-20-02333-f003:**
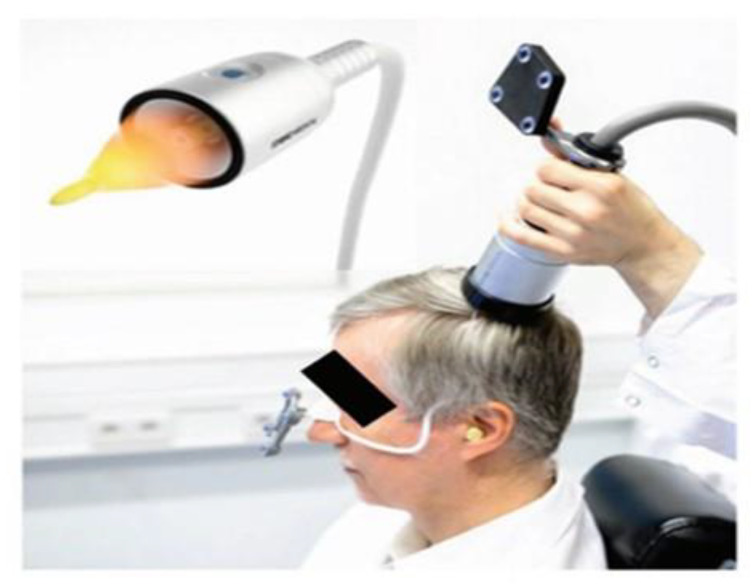
Courtesy image from Beisteiner et al. (2020) [15]. Transcranial pulse stimulation with ultrasound in treating Alzheimer’s disease—a new navigated focal brain therapy. Advanced Science, 7(3), 1902583-N/a. and TPS machine with MRI in QMH, HKU Department of Psychiatry.

**Figure 4 ijerph-20-02333-f004:**
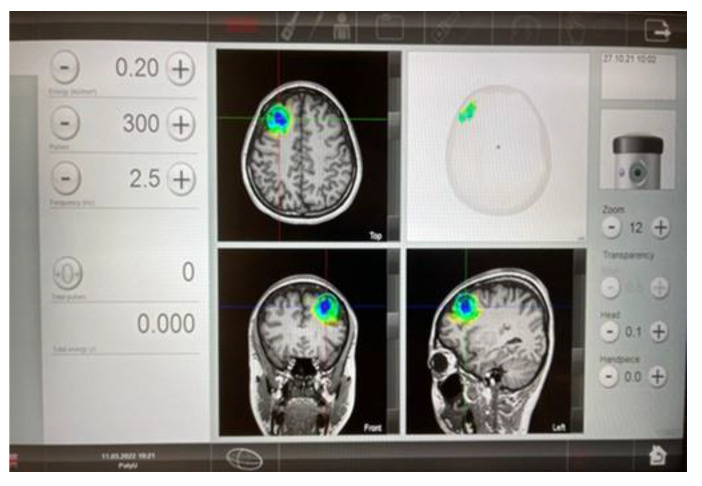
Subject’s fMRI (T1 images) after TPS intervention in our pilot study for MDD.

**Table 1 ijerph-20-02333-t001:** Sociodemographic characteristics between the intervention group (IG) and the waitlist control (WC) group (N = 30).

	IG (n = 15)	WC (n = 15)	
	Mean (SD)/n (%)		*p*
Age	38.8 (15.0)	34.3 (16.5)	0.44
Gender			>0.99
Male	4 (27)	4 (27)	
Female	11 (73)	11 (73)	
Living with family members	2.53 (1.19)	2.8 (1.47)	0.59
Education level			0.1
Elementary or below	1 (7)	1 (7)	
High school	0 (0)	4 (27)	
University or above	14 (93)	10 (67)	
Marital status			0.5
Single	6 (40)	8 (53)	
In a relationship	3 (20)	1 (7)	
Married	5 (33)	6 (40)	
Divorced/separated	1 (7)	0 (0)	
Widowed	0 (0)	0 (0)	
Occupation			0.41
Administrative/clerical staff	1 (7)	0 (0)	
Managerial staff	1 (7)	0 (0)	
Casual worker	0 (0)	1 (7)	
Students (full-time)	5 (33)	6 (40)	
Housewife	0 (0)	2 (13)	
Licensed professionals	3 (20)	1 (7)	
Retirees	0 (0)	1 (7)	
Unemployed	5 (33)	4 (27)	
Income (HKD)			0.16
≤20,000	4 (27)	8 (53)	
>20,000–49,999	4 (27)	5 (33)	
>50,000–79,999	4 (27)	2 (13)	
≥80,000	3 (20)	0 (0)	
Chronic illness			>0.99
Yes	1 (7)	1 (7)	
No	14 (93)	14 (93)	
Psychiatric history (personal)			>0.99
Yes	15 (100)	15 (100)	
No	0 (0)	0 (0)	
Duration of having major depressive disorder (in months)	98 (113)	48.4 (38.3)	0.12
Currently taking prescribed antidepressants			0.08
Yes	9 (60)	14 (93)	
No	6 (40)	1 (7)	
Duration of taking prescribed antidepressants (in months)	33.5 (48.5)	39.1 (34.8)	0.72

**Table 2 ijerph-20-02333-t002:** Effects of the TPS intervention on the depression symptom score (primary outcome) between pre-and post- test (n = 30).

Time Points	Intervention (n = 15)	Control (n = 15)			
	Mean (SD)		Mean Difference	*p*	*d*
Pre-test	25.73 (9.45)	21.60 (8.70)	4.13	0.15	
Post-test	13.20 (7.24)	19.80 (6.89)	−6.60	0.02	−0.93

**Table 3 ijerph-20-02333-t003:** The differences in the secondary outcomes between pre- and post-test (n = 30).

Secondary Outcomes	Baseline	Post-Test	Mean Difference	*p*	*d*
Cognition	26.03 (3.74)	28.7 (1.97)	2.64	0.003	0.88
Trail making test-A	11.22 (8.35)	8.31 (6.05)	−2.91	<0.001	−0.40
Trail making test-B	35.9 (25.40)	33.7 (23.7)	−2.15	0.07	−0.09
IADL	23.0 (5.42)	24.8 (3.46)	1.83	<0.001	0.40
Anhedonia	20.8 (9.02)	14.8 (7.75)	−6.0	<0.001	−0.79
DS_Forward	11.9 (2.40)	12.8 (1.91)	0.93	0.003	0.43
DS_Backward	8.03 (2.67)	10.2 (3.44)	2.14	<0.001	0.69

**Table 4 ijerph-20-02333-t004:** The differences in the primary and secondary outcomes between baseline, post-test, and at the 3-month follow-up (n = 28).

		Baseline (T0)	Post Test (T1)	3-Month Follow-Up (T2)		Post Hoc Test	Effect Size
	n		Mean (SD)		*p*		*d* (T2-T0)
HDRS-17	28	20.9 (7.73)	12.1 (8.03)	11.0 (6.86)	<0.001	T0 > T1 = T2	−1.35
Global cognition	28	26.0 (3.74)	28.7 (1.97)	29.4 (1.42)	<0.001	T0 < T1 = T2	1.20
Trail making test-A	28	11.2 (8.35)	8.31 (6.05)	7.04 (3.89)	<0.001	T0 > T1 = T2	−0.64
Trail making test-B	28	35.9 (25.4)	33.7 (23.7)	7.78 (4.74)	<0.001	T0 = T1 > T2	−1.10
IADL	28	23.0 (5.42)	24.8 (3.46)	25.9 (2.24)	<0.001	T0 < T1 = T2	0.70
SHAPS	28	20.8 (9.02)	14.8 (7.75)	13.2 (9.66)	<0.001	T0 > T1 = T2	−0.81
DS_Forward	28	11.9 (2.40)	12.8 (1.91)	13.0 (1.58)	0.02	T0 < T1 = T2	0.54
DS_Backward	28	8.03 (2.67)	10.2 (3.44)	10.9 (2.62)	<0.001	T0 < T1 = T2	1.09

## Data Availability

The original contributions presented in this study are included in the article/supplementary material, further inquiries can be directed to the corresponding authors.

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
