# Peer review of "Effects of Transcranial Pulse Stimulation (TPS) on Adults with Symptoms of Depression—A Pilot Randomized Controlled Trial"

_ijerph, 2023, doi:10.3390/ijerph20032333_

Round 1

Reviewer 1 Report

The study is highly relevant in the treatment of Depression. The study is well executed, designed and reported. The research problem is properly formulated,  and the aims are stated clearly. The research design is applicable, and the statistical the statistical methods are explained well. The conclusion is is clear. 

Authors, please take note that 17 of your 50 references are older than 5 years. Please update where possible.

Author Response

We have checked all the existing references and cannot find any updates in literature search from 2018-2023 inclusive, as TPS is the latest technological device for treatment of depression. Thank you.

Reviewer 2 Report

I appreciated both original topic and methodology. 

With regard to the "results", table n.1 and its discussion are too large and dispersive, It would be better to revise that.

I suggest to pay more attention to APA standards on both English and References.

I kindly recomend to revise whole article in order to correct spelling at the end of many lines (17, 19, 26, 27, 36, 37 and so on until References' paragraph)

Author Response

  • We have attempted to revise those tables and discussion part but we cannot trim down any components in these sections, given the fact that this is the first pilot RCT nationwide, and readers should be able to read a more comprehensive context regarding this intervention. We thank you reviewer’s comments regardless.
  • This paper has been edited in November 2022 (see attachment) and references are using the IJERPH guidelines. 
  • The whole manuscript has been cross-checked again. Thank you.

Reviewer 3 Report

Dear authors,

I am grateful to have been able to read your article, which I found very interesting. I congratulate you on your research.

Here are some suggestions that may help to improve the article.

You talk about the lack of similar studies in your country, but are there studies in other countries?

You include a sample of young adults, but instead the age ranges up to 54. What do you mean by young adult?

I consider the sample to be small even for a pilot study. At 3 months it is less than 30, which makes statistical reliability difficult.

Methodologically it is not a clinical trial, as the initial selection is not randomised but by volunteers.

It is not explained how they ensured that patients met the inclusion criteria. 

Exclusion criteria that the author should suspect to be related to the therapy are detailed, but the introduction does not identify these risks.

The way the treatment process is described, it seems that everything was done by the two doctors alone. Is this the case, and did they take care of contacting patients to remind them of appointments?

I consider it unethical to provide a voucher for a supermarket to the participants. Did the ethics committee accept this point? It would be good to attach the certificate of the ethics committee if this has not already been done.

Is the proportion of women in the sample representative of the prevalence of major depression in women?

The limitations are not in the study but in the applicability of the technique.

The conclusions are very risky in generalising the benefit of this treatment only tested in 28 people.

I hope you find my contributions helpful.

The reviewer.

Round 2

Reviewer 3 Report

Dear authors,

unfortunately I do not believe that most of the indications have been taken into account.

Perhaps what limits the study the most, even though it is a pilot study, is the sample loss. Before starting the study, the losses in it must be counted and more participants must be selected so that the minimum is always 30.

The randomization of the sample is not real. There is no database from which patients are selected. They are voluntary patients who are then randomized.

These limitations, together with the errors that have not been adequately answered, lead me to assess the study negatively and wish them luck in the editorial process in this or another journal.

For greater success with your article, I recommend following the reviewers' instructions.

Best regards,

The reviewer

Author Response

In this study, all applicants had to complete a QR code Google online application form to apply for participation. After checking applicants’ sociodemographic information including gender/age/year of diagnosis and current medication etc according to the inclusion/exclusion criteria, eligible participants received a screening survey (HAM-D-17) to assess the level of depression severity. Those eligible subjects were interviewed separately via zoom meetings by the Principal investigator and the research associates to ensure that all participants understood the nature and purpose of the study. Our estimated sample size only required 30 subjects in this study and all the subjects had completed all the TPS intervention, and all subjects completed the post test immediately after intervention and at 1-month. Only 2 subjects suddenly attrited from the 3-month follow-up as they relocated and could not be traced.The project team cannot recruit another 2 subjects to replace the 2 attrited subjects as the recruitment stage was over and could not be extended due to tight budget and time constraints. Ideally, we could consider attrition rates, say, 20% in intervention study, but we were limited by very tight budget since each participant had to undertake pre-and post MRI scanning which took a lion share out of a small internal funding. We would consider this attrition as one of the limitations in this regard but since this is a pilot feasibility study, we recommend that future replication of this study should include larger representative samples. Thank you for editor’s understanding and reviewer’s feedback.